# The Potential Role of the Methionine Aminopeptidase Gene *PxMetAP1* in a Cosmopolitan Pest for *Bacillus thuringiensis* Toxin Tolerance

**DOI:** 10.3390/ijms232113005

**Published:** 2022-10-27

**Authors:** Min Ye, Lei Xiong, Yi Dong, Chao Xie, Zhen Zhang, Lingling Shen, Zeyun Li, Zhen Yue, Puzi Jiang, Zhiguang Yuchi, Minsheng You, Shijun You

**Affiliations:** 1State Key Laboratory for Ecological Pest Control of Fujian and Taiwan Crops, Institute of Applied Ecology, Fujian Agriculture and Forestry University, Fuzhou 350002, China; 2International Joint Research Laboratory of Ecological Pest Control, Ministry of Education, Fujian Agriculture and Forestry University, Fuzhou 350002, China; 3Ministerial and Provincial Joint Innovation Centre for Safety Production of Cross-Strait Crops, Fujian Agriculture and Forestry University, Fuzhou 350002, China; 4Key Laboratory of Integrated Pest Management for Fujian-Taiwan Crops, Ministry of Agriculture and Rural Affairs, Fuzhou 350002, China; 5BGI-Sanya, BGI-Shenzhen, Sanya 572025, China; 6School of Pharmaceutical Science and Technology, Tianjin University, Tianjin 300072, China

**Keywords:** *Plutella xylostella*, methionine aminopeptidase, Bt tolerance, genetic linkage, CRISPR/Cas9

## Abstract

Methionine aminopeptidases (MetAPs) catalyze the cleavage of the N-terminal initiator methionine (iMet) in new peptide chains and arylamides, which is essential for protein and peptide synthesis. MetAP is differentially expressed in two diamondback moth (DBM; *Plutella xylostella*) strains: the G88 susceptible strain and the Cry1S1000 strain, which are resistant to the Bt toxin Cry1Ac, implicating that MetAP expression might be associated with Bt resistance. In this study, we identified and cloned a MetAP gene from DBMs, named *PxMetAP1*, which has a CDS of 1140 bp and encodes a 379 amino acid protein. The relative expression of *PxMetAP1* was found to be ~2.2-fold lower in the Cry1S1000 strain compared to that in the G88 strain. *PxMetAP1* presents a stage- and tissue-specific expression pattern, with higher levels in the eggs, adults, integument, and fatbody of DBMs. The linkage between *PxMetAP1* and Cry1Ac resistance is verified by genetic linkage analysis. The knockout of *PxMetAP1* in G88 by CRISPR/Cas9 leads to a ~5.6-fold decrease in sensitivity to the Cry1Ac toxin, further supporting the association between the *PxMetAP1* gene and Bt tolerance. Our research sheds light on the role of MetAP genes in the development of Bt tolerance in *P. xylostella* and enriches the knowledge for the management of such a cosmopolitan pest.

## 1. Introduction

*Bacillus thuringiensis* (Bt) is an efficient entomopathogenic micro-organism that has been researched indepth and is extensively used at present [1]. It was first made into a wettable powder in order to kill Lepidoptera pests through the spraying of spores [2]. Most organisms are unaffected by the Bt Cry1Ac toxin, making it an environmentally friendly insecticide [3]. Thus, the Cry1Ac toxin is frequently used as a bioinsecticide in a wide range of commercial applications, as well as being expressed in transgenic plants for pest control [4,5,6,7,8]. However, the continuous and large-scale use of various Bt formulations has resulted in increased resistance in the target pests (to Bt) under continuous selection. In 1985, McGaughey first reported that *Plodia interpunctella* developed resistance to Bt in the laboratory [9]. Five years later, Tabashnik et al. (1990) determined the resistance of *Plutella xylostella* to Bt for the first time in a field where Bt formulations had been used for a long time [10], and the cumulative actual resistance cases of transgenic crops to Bt toxin grew from 3 cases in 2005 to 16 cases in 2016 [11]. Many hypotheses concerning the action mechanism of the Bt toxin have been proposed, including a continuous binding model [12], signal transduction model [13], direct-acting models [14], and dual-acting models [15]. The continuous binding model believes that the Cry-activated toxin first binds to the highly abundant GPI-anchored receptor protein with a lower binding force, forming a preporous oligomeric structure. The preporous oligomers then bind to other receptors with a higher binding force, eventually inserting into the intestinal membrane, causing perforation, and resulting in the death of the insect [16,17,18]. The signal transduction model suggests that the Cry-activating toxin binds to cadherin to activate G protein and adenylate cyclase, promote the increase of intracellular cAMP levels, and activate protein kinase A-related intracellular death signaling pathways [13]. However, Cry protoxins are also likely to bind to cadherin and oligomerize under the action of intestinal proteases, resulting in the death of the worm [19]. In addition, the dual-action model proposed that Cry protoxins can directly bind to cadherin without protease activation and oligomerize to form a preporous oligomer different from the activated protein and then interact with the receptor. The body molecules combine and insert into the intestinal membrane, causing the membrane to perforate, resulting in the death of the insect [20]. Current research reports on the mechanism of Bt resistance in lepidopteran insects mainly involve the down-expression of trypsin, which reduces protoxin conversion to activated toxin [21,22,23,24], reduces the binding ability of the Bt toxin to midgut receptors through reducing the activity and transcription of alkaline phosphatase (ALP) or aminopeptidase N (APN), as well as mutations of APN, cadherin-like (CAD), and ATP-binding cassette (ABC) transporters [17,25,26,27,28,29,30,31], isolates the toxin by glycolipid or esterase, reducing the probability of toxin receptor binding [32,33], triggers the defense mechanism in the midgut by enhancing the immune response to increase body resistance [34], etc. However, the formation of resistance is a very complex process, as the resistance of insects to Bt may occur in every link of toxin action.

Methionine aminopeptidases (MetAPs), also known as peptidase M, are a class of divalent metal ion (Co^2+^)-dependent proteases that exist in the cytoplasm and organelles [35]. They catalyze the cleavage of the N-terminal initiator methionine (iMet) in new peptide chains and arylamides (called N-terminal methionine excision (NME)), which is essential for protein and peptide synthesis [36]. NME is versatile and essential for the post-translational processing and homeostasis of newly synthesized proteins and peptides [37,38], as well as in maintaining the normal physiological functions of cells [39]. It has been found that there are differences in the expression of proteins involved in lipid metabolism, cytoskeleton growth, cell proliferation, and protein synthesis when the activity of MetAPs is disturbed [40].

MetAP1 and MetAP2 are two types of MetAPs found in eukaryotes [41,42]. MetAP1 is responsible for overseeing the co-translational excision of the first methionine residue in eukaryotes. The loss of function of this gene may lead to translation defects in many essential proteins in the cells [43], and the retention level of iMet is negatively correlated with the cell proliferation rate. Compared with MetAP1, MetAP2 contains a unique Cys (228)–Cys (448) disulfide bond, which exists in the recombinant enzyme in oxidation and reduction states. It has been found that the allosteric disulfide bond regulates the translation of MetAP2 and controls the substrate specificity and catalytic efficiency [44]. As the catalytic activity of MetAP2 is pharmacologically related to cell growth, angiogenesis, and tumor research [45,46], it is often used as a therapeutic target for various diseases in medicine [47,48,49]. Previous studies have found that MetAPs are also involved in the activation and transport of antibiotics for toxin receptors or at target sites [43,50]. For example, MetAPs have been shown to prevent the growth of *Escherichia coli* and *Staphylococcus aureus* cells with remarkable effectiveness, demonstrating that MetAPs are intracellular targets of *E. coli* [47]. In insects, MetAPs in honeybee cells are inhibited by fumagillin, as a target site and interfere with the protein modifications necessary for normal cell function, changing the structure of honeybee midgut tissue and metabolic proteins, thereby controlling honeybee infection with sporozoites disease and increasing the body’s resistance [51].

*P. xylostella* is a major pest that seriously harms vegetables all over the world and is resistant to many insecticides. The insecticide resistance of *P. xylostella* has become a focal issue for researchers in recent decades, and the search for highly efficient, specialized, and environmentally friendly methods for biological control is an important trend for future development. The resistance mechanisms of *P. xylostella* to Bt toxins mainly involve dissolution and the incomplete processing of the protoxin [52], binding site alterations [53], differential gene expression [54], toxin sequestration or condensation [55,56], immune response alterations [57], and physiological adaptations [58], among other aspects. In recent years, it has also been found that the mitogen-activated protein kinase (MAPK) signaling pathway can trans-regulate gene expression differences, resulting in *P. xylostella* developing significant levels of Bt resistance [59]. The MAPK cascade was stimulated and regulated by some hormones, resulting in the resistance of *P. xylostella* to the Bt Cry1Ac toxin by affecting the expression levels of ALP and ABCC genes in the midgut [60]. In subsequent studies, p38, ERK, GATAd, and POUM1 are also confirmed to be involved in the MAPK signaling pathway to a large extent by regulating the differential expression of multiple Cry toxin receptors and their nonreceptor analogs in the midgut of *P. xylostella* [61,62,63]. At the same time, further research on the three potential activation pathways of the complex four-layer MAPK signaling module provides insights into better control of *P. xylostella* [62].

Bt Cry toxins primarily act on the midgut of insects [12], but it is not clear whether genes with a different expression before and after the action of Bt toxins are expressed in the midgut tissue exclusively, and the roles of the differentially expressed genes in the resistance of *P. xylostella* to Bt need to be further verified. Based on the midgut transcriptome of the Geneva 88 (G88)-susceptible strain and the Cry1S1000-resistant strain (which is resistant to the Cry1Ac toxin) [64], 11 differentially expressed aminopeptidase genes in *P. xylostella* are potentially linked to the resistance of the Bt toxin to Cry1Ac. There are 5 out of the 11 genes that have not yet been investigated. In this study, we focus on *PxMetAP1* (*Px013243*), the first one that was successfully amplified, to explore its potential role in Bt toxin tolerance. Our results demonstrate that the alternative splicing form of the *PxMetAP1* at the gene level is linked to Cry1Ac tolerance, and the knockout of *PxMetAP1* increases the tolerance of *P. xylostella* to multiple Bt toxins. The findings of this study enrich the mechanisms underlying *P. xylostella*’s tolerance to Bt toxins.

## 2. Results

### 2.1. Differential Expression of PxMetAP1 Gene in the G88 and Cry1S1000 Strains

A significant difference in the relative expression of the *PxMetAP1* gene in the midgut tissue of *P. xylostella* 4th instar larvae was identified between the G88 and Cry1S1000 strains, i.e., the relative expression of *PxMetAP1* gene in the midgut tissue of *P. xylostella* 4th instar larvae was ~2.2-fold lower in the Cry1S1000 strain than in the G88 strain (independent samples *t*-test, *p* < 0.05) (Figure 1). This result is consistent with previous transcriptome data [64].

### 2.2. Identification and Sequence Characteristics of the PxMetAP1 Gene

The CDS of the cloned *PxMetAP1* gene (GenBank accession no. ON813083) was 1140 bp in length and encoded 379 amino acids (Figure 2a). *PxMetAP1* belongs to the PLN03158 superfamily and is defined as methionine aminopeptidases, according to conservative domain prediction (Figure 2b). The analysis of its sequence characteristics showed that the protein has a molecular weight of 41.809 kDa and a theoretical isoelectric point of 6.22. It is an unstable hydrophilic protein lacking a transmembrane structure and signal peptide but does possess one N-glycosylation site and five O-glycosylation sites. *PxMetAP1* has 11 motifs in insects and is highly conserved at five sites (Figure 2c).

### 2.3. Expression Pattern of PxMetAP1 Gene

The relative expression of the *PxMetAP1* gene in different developmental stages and tissues was detected by qPCR. The relative expression of the *PxMetAP1* gene was highest in eggs and adults, followed by the 1st and 4th instar larvae, and was lowest in the 3rd instar larvae of various developmental stages (LSD test, *p* < 0.05; Figure 3a). In terms of different tissues, the relative expression levels were highest in the integument and fatbody, followed by the head, and the lowest in the midgut and Malpighian tubule (LSD test, *p* < 0.05; Figure 3b).

### 2.4. Genetic Linkage of PxMetAP1 Gene with Cry1Ac Resistance

In order to verify the relationship between the *PxMetAP1* gene and Cry1Ac resistance of *P. xylostella*, the DNA fragment of the *PxMetAP1* gene containing only the intron 3 region was cloned with specific primers, as the polymorphism of this intron enables PCR to amplify different intron lengths of the *PxMetAP1* alleles (the length of intron 3 in G88 and Cry1S1000 strains is 464 bp and 268 bp, respectively, and the respective genotypes are *S_M1_S_M1_* and *R_M1_R_M1_*: Figure 4a) and, so, this feature can be used to identify the genotype of *PxMetAP1*.

The discriminating concentration of the Cry1Ac toxin in the toxicity bioassay that could kill 100% of F_1_ heterozygotes was 0.5 μg/mL, and the toxicity bioassay of 10 back-cross families was tested using this concentration of the Cry1Ac toxin (Figure 4b). The genotypes of all surviving individuals in each back-cross family were identified based on *S_M1_* and *R_M1_* as allelic tags, and G88 and Cry1S1000 were found to have homozygous *S_M1_* and *R_M1_* alleles, which hybridized to obtain F_1_ that was heterozygous for the *R_M1_S_M1_* genotype (Figure 4c). The results showed that 206 of the 400 larvae in the treatment group survived after toxin treatment, and all 417 larvae in the control group survived (Appendix A). The genotype of the surviving individuals in the treatment group of each back-cross family was *R_M1_R_M1_*, and the proportion was significantly higher in the treatment group than in the control group (Fisher’s exact test, *p* < 0.001 for each family); there was no significant difference between the expected 50% proportion of genotypes with *R_M1_R_M1_* versus *R_M1_S_M1_* in the controls (Fisher’s exact test, *p* > 0.44 for each family; Figure 4c, Appendix A). These results indicate that the *PxMetAP1* genotype is related to the survival rate of Cry1Ac toxin treatment, showing that the *PxMetAP1* of *P. xylostella* has a genetic linkage with Cry1Ac resistance.

### 2.5. CRISPR/Cas9-Mediated Knockout of PxMetAP1

In order to improve the efficiency of *PxMetAP1* gene knockout, one target was designed for each of the sense and antisense strands of exon 2, respectively, and the Cas9 protein and sgRNA of the two targets were proportionally added and mixed for incubation during microinjection. A wild-type strain of G88 was injected with 140 eggs, and mutation detection revealed successful editing of the sgRNA1 locus.

The single-pair cross of the G_0_ mutant individuals with the G88 wild strain individuals was carried out to generate F_1_. F_1_ continued to single-pair cross with G88 individuals as they developed to adulthood, and mutation detection was continued by PCR amplification. The progeny of individuals with nested peaks in the detection results were retained and raised to adults. After that, the method of single mating was used to retain the progeny of those individuals with nested peaks in the detection results until the homozygous mutant strain was screened. A homozygous mutant strain with a 7 bp insertion was obtained when *PxMetAP1* was screened to F_4_, named PxMetAP1-KI-7 (Figure 5), which provided a stable genetic test object for subsequent functional analysis.

### 2.6. Effects of PxMetAP1 Knockout on the Activity of MetAPs

As shown in Figure 6a, the standard curve equation of PNA was Y = 0.0086X − 0.00189, where R^2^ = 0.9999 (Y represents the value and X represents the actual PNA concentration). The real-time concentration of PNA was calculated according to the standard curve equation, and the activity of the MetAPs was calculated by substituting it into the enzyme activity formula, thus obtaining the MetAP activity assay results of 17.225 IU/mL and 12.526 IU/mL for G88 and PxMetAP1-KI-7, respectively. The analysis showed that PxMetAP1-KI-7 had considerably lower MetAPs activity than G88 (independent sample *t*-test, *p* < 0.001; Figure 6b), indicating that the knockout of the *PxMetAP1* gene reduced the MetAP activity in *P. xylostella*, thus suggesting that the gene might have lost its original function.

### 2.7. Effects of PxMetAP1 Knockout on the Sensitivity of Bt Toxin

The bioassays indicated that the sensitivity of the *PxMetAP1* mutant homozygous strain to the Cry1Ac, Cry1Ab, Cry2Aa, and Cry2Ab toxins decreased by 5.58-, 3.45-, 3.07-, and 2.26-fold, respectively, compared with the G88 susceptible strain. The sensitivity of the G88-No-Indel strain to Cry1Ac was basically the same as that of the G88 strain (Figure 7 and Appendix A). These results suggest that deleting the *PxMetAP1* gene could increase Bt toxin tolerance in *P. xylostella*, which might be involved in a certain link of the resistance/tolerance regulation pathway.

## 3. Discussion

In this research, three lines of evidence were obtained which indicate that the MetAPs gene *PxMetAP1* is involved in mediating the tolerance of *P. xylostella* to Bt toxins. First, the midgut expression level of *PxMetAP1* was significantly down-regulated in the Cry1S1000-resistant strain. Second, genetic linkage experiments proved that the intron polymorphism of *PxMetAP1* at the gene level was closely related to Bt Cry1Ac toxin resistance. Finally, the deletion of *PxMetAP1* using CRISPR/Cas9 gene-editing technology increased the tolerance of *P. xylostella* to Bt toxins. In conclusion, the above results demonstrate that the *PxMetAP1* gene is involved in the process of Bt toxin tolerance in *P. xylostella*.

MetAPs are an evolved class of enzymes that cleave iMet residues on 60–70% of nascent peptide chains in all living cells [65]. This is a fundamental and universal protein modification that is a major source of N-terminal protein form diversity [36]. It is essential for cellular maturation, growth, and defense, as well as maintaining cellular homeostasis [38]; in addition, it is involved in the activation and transport of antibiotics as transcriptional regulators, site-specific recombination factors, and toxin receptors [43]. Although there is usually only one MetAPs gene in prokaryotic genomes, eukaryotes express two types of MetAPs, which can be classified into type 1 and type 2 based on sequence differences [40]. A study has discovered that nearly all type 1 MetAPs have a cysteine at the top of the active region, which is crucial for recognizing methionine as a substrate, potentially affecting major parts of the protein through post-translational modifications [65]. Although MetAP2 is known to be involved in the development of specific tissues in multi-cellular organisms, the physiological role of MetAP1 has not been thoroughly studied [66,67], with reported results primarily as an antimicrobial target site [68]. A recent study has shown that certain proteins involved in lipid metabolism, cytoskeletal tissue, cell proliferation, and protein synthesis are differentially expressed in the protein group and transcriptome when MetAP1 activity is disrupted [69]. Although the MetAP1 gene serves a number of functions, its complete functions remain unknown. In this aspect, a new effect of this traditional multifunctional gene was reported in terms of its relationship with Bt tolerance in insects within this study.

The *PxMetAP1* gene presented stage- and tissue-specific expression in *P. xylostella*, with the highest relative expression levels in the egg and adult stages, followed by 1st and 4th instar larvae, with the lowest relative expression level in 3rd instar larvae. The 4th instar is in the prophase of pupation, which is an important stage of growth and development, with enhanced physiological and biochemical activities (e.g., food intake [70]). Higher expression of Methionine aminopeptidase, known to play an important role in maintaining normal cell function [39], in the 4th instar, therefore, is potentially linked with the growth and development of *P. xylostella*. The highest relative expression of the *PxMetAP1* gene in the egg and adult stages may be related to its mediating processes, such as cell proliferation and skeletal tissue development [40]. Despite the difference in the expression abundance of this gene in different developmental stages, it was found to be expressed throughout the growth and developmental stages. Therefore, this gene may play a key role in processing polypeptides and synthesizing new proteins, thus indirectly mediating many physiological processes in *P. xylostella*. Among different tissues, the relative expression of the *PxMetAP1* gene was highest in the integument and fatbody, possibly because MetAPs are related to the development of specific tissues [71]. Related research has shown that the integument and fatbody of insects can produce immunity and detoxification to exogenous harmful substances [72,73], thereby enhancing their resistance to external adverse factors. The relative expression levels in the midgut and Malpighian tubule were the lowest; although Bt toxins mainly function on the midgut of larvae, some Bt toxin receptor genes or genes involved in Bt insecticidal process could also be significantly expressed in other tissues [74,75,76].

Up to now, it has been found that the resistance mechanism of insects to Cry toxins is possibly related to changes in the binding of Cry proteins to midgut receptors, changes in protoxin processing, enhanced immunity, toxin sequestration, epithelial regeneration, as well as amino acid replacement, etc. [77]. In recent years, it has also been found that multiple factors in the MAPK signaling pathway transregulate the differential expression of midgut genes to promote insect resistance to Cry toxins [61,62,63,78]. It has been widely accepted that the resistance of insects to Cry toxin will be regulated in many ways, and the potential mechanism may be an extremely complex network worthy of further investigation. Although the function of the MetAP1 gene has been reported in mammals, its role in Bt biopesticide resistance/tolerance in insects has been poorly analyzed. MetAPs are considered to be the core component of cells, thus potentially representing a highly essential protein [79]. We successfully constructed a homozygous mutant strain of the *PxMetAP1* gene using CRISPR/Cas9 technology, having a frameshift mutation in exon 2. The definition of insecticide resistance level is defined by resistance factors (RFs), based on previous studies [80,81]: susceptibility (RF = 1), decreased susceptibility (RF = 3–5), low resistance (RF = 5–10), moderate resistance (RF = 10–40), high resistance (RF = 40–160), and very high resistance (RF > 160). In this study, we found that the knockout of the *PxMetAP1* gene resulted in tolerance to Cry1Ac toxin and decreased susceptibility to Cry1Ab and Cry2Aa toxins. 

Previous studies have shown that most lepidopteran insects, which are resistant to Cry1A toxins, are likely to be susceptible to Cry2A toxins, and vice versa, since Cry1A and Cry2A toxins do not share binding sites [82,83,84,85,86,87]. Exceptions, however, have been reported as well. For example, the mutations of different alleles of the cadherin gene *PgCad1* in *Pectinophora gossypiella* showed cross-resistance to Cry1Ac and Cry2Ab, (i) the strain with homozygous allele *r14* develops a 237-fold resistance to Cry1Ac and a 1.8-fold cross-resistance to Cry2Ab [88], (ii) the strain with the homozygous allele *r16* shows a 300-fold resistance to Cry1Ac and a 2.6-fold cross-resistance to Cry2Ab [89], and (iii) the strain with the homozygous allele *r13* has a 220-fold resistance to Cry1Ac and a 2.1-fold cross-resistance to Cry2Ab [90]. In addition, after screening cotton bollworm (*Helicoverpa armigera*) with Cry1Ac for more than 200 generations in a laboratory, it produced up to a 1000-fold resistance to Cry1Ac and a 6.8-fold cross-resistance to Cry2Ab, whereas multiple selections with Cry2Ab yielded a 5.6-fold resistance as well as a 61-fold cross-resistance to Cry1Ac [91]. Comparable phenotypes have also been identified in *Heliothis virescens*, e.g., Jurat-Fuente et al. (2003) found that the laboratory *H. virescens* strain CP73-3 was cross-resistant to both Cry1A and Cry2A [92]. Moreover, transgenic cotton, which produces the Bt toxins Cry1Ac and Cry2Ab, is widely used to control lepidopteran pests, as it is generally accepted that cross-resistance between these toxins is unusual. However, this may not be the case sometimes. The Cotton pest *P. gossypiella*, which was selected by Cry2Ab in the laboratory, produced a 240-fold resistance to Cry2Ab and a 420-fold cross-resistance to Cry1Ac [93]. Besides, two strains of *H. armigera* with significant resistance to Bt Cry1Ac found in cotton fields (460- and 1200-fold) showed cross-resistance to Cry2Ab (4.2- and 5.9-fold) [94]. Brévault et al. (2013) also reported that eight main lepidopteran pests commonly developed cross-resistance to Cry1A and Cry2A toxins from transgenic cotton [95]. An unusual but interesting result was presented recently by Zhong et al. (2022) that the midgut cadherin (CmCad) of *Cnaphalocrocis medinalis* shared Cry1Ac and Cry2Aa binding sites; the knockout strain of *CmCad1* simultaneously showed resistance to both Cry1A and Cry2A toxins [96].

Cadherins, like aminopeptidases, are well-known as Bt toxin receptors. However, studies have revealed that cadherin can also be used as a novel synergist to enhance the toxicity of insect larvae in response to a variety of Cry toxins [97,98,99,100]. The emergence of Bt synergists can play a role in correcting the deficiencies of Bt biopesticides [101]. The use of such synergists not only allows for lower doses of Bt and a wider range of activity for pest control [102,103] but also delays the development of resistance in the target pest [104]. It has been found that cadherin can act as a synergist for Bt toxins against a variety of lepidopteran insects; for example, the cadherin fragment synergistically enhanced the virulence effect of the Cry1A toxin against *Agrotis ipsilon* and *Spodoptera exigua*, while also enhancing the toxicity of the Cry1C toxin against *S*. *exigua* [105]. Later, Rahman et al. (2012) found that cadherin orthologs also increased the toxicity of the Cry1Fa toxin in *Spodoptera frugiperda* larvae to varying degrees [106]. In addition, cadherin, together with Cry1Ac toxin, showed increased lethality in *H. armigera* and *P. xylostella*, where it was found that the synergistic effect in *P. xylostella* was stronger [107,108]. As a novel synergist, cadherin also enhanced larval toxicity in combination with the Cry4Ba toxin in Diptera *Aedes aegypti* [109] as well as with Cry3Aa, Cry3Bb, and Cry8Ca in Coleopteran *Alphitobius diaperinus* [110]. Therefore, combined with the biological function of MetAP1 (in initiating polypeptide synthesis and the phenomenon wherein it plays a role in mediating the tolerance of *P. xylostella* observed in this study), developing it as a novel synergist candidate may broaden the toxicity range of Bt toxins to other important agricultural lepidopteran pests, thus increasing its use in agriculture. 

In conclusion, our findings imply that the *PxMetAP1* gene is implicated in the tolerance of *P. xylostella* to Bt toxins. The polymorphism of its intron was found to be closely associated with tolerance at the gene level, but it remains unclear how the gene subsequently mediates the generation of tolerance. The generation of Bt resistance/tolerance is a highly complex process influenced by the interaction of numerous factors. Therefore, further verification (e.g., through omics sequencing) should be carried out in order to explore the physiological effect of the deletion of P*xMetAP1* in *P. xylostella*, revealing the substrate spectrum of *PxMetAP1* in vivo at the transcription and protein level and searching for more genes linked to Bt resistance/tolerance, which are expected to help in elucidating the potential Bt resistance mechanism of *P. xylostella,* and aid in the design of more focused management measures effective in delaying the development of Bt resistance in insect populations.

## 4. Materials and Methods

### 4.1. Insect Strains

The G88 susceptible strain of *P. xylostella* used in this experiment was kindly provided by Dr. Anthony M. Shelton of Cornell University [111], while the Cry1S1000 resistant strain was a stable and homozygous-resistant population established after the multigeneration screening of the Cry1Ac-R strain [111], with a concentration of 1000 μg/mL using Cry1Ac toxin in the early stages [112]. Toxicity bioassays showed that the concentration (LC_50_) of the Cry1Ac toxin for the Cry1S1000 strain was higher than 1000 μg/mL, and the resistance of the Cry1S1000 strain to Cry1Ac was over 8000-fold (to the G88 strain) [112]. The rearing conditions for all strains were 26 ± 0.5 °C, 60 ± 5% RH (relative humidity), and a photoperiod of 16 h:8 h (light:dark). The larvae of *P. xylostella* were fed an artificial diet, while the adults were fed 10% honey water. The homozygous strain of *PxMetAP1* mutation of *P. xylostella*, selected after CRISPR/Cas9 knockout, was raised in the same way as the above two strains.

### 4.2. RNA Extraction and cDNA Synthesis

Total RNA of 4th instar larvae of *P. xylostella* G88 and Cry1S1000 strains were extracted using an Eastep^TM^ Super Total RNA Extraction Kit (Promega, Beijing, China), and the concentrations were determined using a micro analyzer (Nanodrop 2000, Thermo Fisher, Waltham, MA, USA). Furthermore, the extraction quality was detected by 2% TAE agarose gel electrophoresis. The cDNA was produced through a reverse transcription reaction with FastKing gDNA Dispelling RT SuperMix (Tiangen, Beijing, China), and the reaction product was stored at −20 °C.

### 4.3. Real-Time Quantitative PCR (qPCR) Analysis

Real-time quantitative PCR (qPCR) is often used to analyze the relative expression of genes. The qPCR primer for the *PxMetAP1* gene was designed using the primer design tool (https://www.ncbi.nlm.nih.gov/tools/primer-blast/, accessed on 29 September 2022) on the NCBI website, and *RPL32* (ribosomal protein L32, GenBank accession no. AB180441) was used as the internal reference gene. The primer sequence is provided in Table 1. The samples of *P. xylostella* at different developmental stages were collected, including seven groups of eggs, 1st to 4th instar larvae, pupae, and adults. The samples of different tissues included five groups (head, integument, midgut, fatbody, and Malpighian tubule), which were obtained by dissecting the 4th instar larvae. The obtained samples were promptly frozen in liquid nitrogen and stored at −80 °C for RNA extraction. The steps for RNA extraction and cDNA synthesis are described in Section 4.3. The experiment was conducted according to the instructions of Eastep^®^ qPCR Master Mix (Promega, Beijing, China), with four biological replicates for each treatment and three technical repetitions for each biological replicate. The 2^−ΔΔCt^ method was used for the relative quantification of the data [113], and the results were analyzed by one-way ANOVA using the SPSS Statistics 22 software (v22.0, Norman H. Nie, C. Hadlai (Tex) Hull and Dale H. Bent, Chicago, IL, USA) in order to determine the significant differences among different treatment groups, while the LSD method was used for multiple comparisons (*p* < 0.05).

### 4.4. Gene Cloning and Bioinformatic Analysis

Based on a comparison between the transcriptome data of *P. xylostella* [64] and the genome database of *P. xylostella* (http://59.79.254.1/DBM/index.php, accessed on 24 September 2020), the *PxMetAP1* gene sequence was obtained, and the specific primers were designed using the SnapGene 3.2.1 software. The CDS sequence of *PxMetAP1* was amplified in three segments, then combined into a complete CDS sequence by fusion PCR. PCR amplification was performed using Hieff Canace^®^ High-Fidelity DNA Polymerase, following the manufacturer’s instructions (Yeasen, Shanghai, China). The primers are detailed in Table 1. The PCR products were purified using an EZNA Gel Extraction Kit (Omega, Morgan Hill, GA, USA) and then ligated into pESI-Blunt simple vectors using a Hieff Clone^TM^ Zero TOPO-Blunt Simple Cloning Kit (Yeasen, Shanghai, China). The positive cloning vectors were sequenced by Biosune Biotech Company (Biosune, Fuzhou, China).

The returned sequencing sequences were compared with nucleotide homology using the Blast tool (https://blast.ncbi.nlm.nih.gov/Blast.cgi, accessed on 11 December 2020) on the NCBI website, with the CD Search tool (https://www.ncbi.nlm.nih.gov/Structure/cdd/wrpsb.cgi, accessed on 11 December 2020) to identify the conservative domain (CDD) and the ORF Finder tool (https://www.ncbi.nlm.nih.gov/orffinder/, accessed on 11 December 2020) to identify the open reading frame (ORF). The Translation tool (https://web.expasy.org/translate/, accessed on 26 January 2021) on the ExPASy website was used to convert the nucleotide sequence into a protein sequence; parameters, such as molecular weight, isoelectric point, and instability index, were analyzed using the ProtParam tool (https://web.expasy.org/protparam/, accessed on 26 January 2021), and the protein hydrophobicity was analyzed using the Protscale tool (https://web.expasy.org/protparam/, accessed on 26 January 2021). In the Services and Products section of the DTU Health Tech website, the TMHMM-2.0 tool (https://services.healthtech.dtu.dk/service.php?TMHMM-2.0, accessed on 26 January 2021) was used to predict the transmembrane structure in the proteins, the SignalP-5.0 tool (https://services.healthtech.dtu.dk/service.php?SignalP-5.0, accessed on 26 January 2021) was used to predict the signal peptide in proteins, and the NetNGlyc-1.0 tool (https://services.healthtech.dtu.dk/service.php?NetNGlyc-1.0, accessed on 26 January 2021) and the NetOGlyc-4.0 tool (https://services.healthtech.dtu.dk/service.php?NetOGlyc-4.0, accessed on 26 January 2021) was used to predict the N- and O- glycosylation sites in protein sequences, respectively. Conserved motifs were analyzed using the MEME tool [114] (https://meme-suite.org/meme/tools/meme, accessed on 9 May 2022).

### 4.5. Toxin Synthesis and Toxicity Bioassays

A method previously described in the literature was used to express, extract, and purify the Cry1Ac toxin from Btk strain HD-73 [115,116]. The concentrations of the Cry1Ac, Cry1Ab (Genralpest, Beijing, China), Cry2Aa (Genralpest, Beijing, China), and Cry2Ab (Genralpest, Beijing, China) toxins were determined using the BCA method (Solarbio, Beijing, China), and their purities were analyzed by 10% SDS-PAGE. After this, they were stored at −80 °C until further use.

Four Bt toxins were individually tested for their toxicity against different strains of *P. xylostella* by a residual film method [112], as previously mentioned. Seven concentration gradients were set for each toxin, four biological replicates were set for each concentration, and 15 larvae were picked from each replicate, for a total of 60 larvae. The preheated artificial diet (approximately 5 mL) was poured into a 25 mL round-bottomed plastic cup, after which 200 µL of the toxin solution was spread evenly over the surface of the diet. Fifteen 3rd instar larvae at the early stage were placed in the cup after the surface of the diet was dried. The number of larval deaths was recorded after 72 h, and the mortality rate was corrected using Abbott’s formula [117]. The resistance ratio (RR) was calculated by dividing the LC_50_ (half lethal concentration) of each strain by the LC_50_ of the G88 strain using the probit tool in the SPSS Statistics 22 software. Two strains were considered to be significantly different in sensitivity to Bt toxin if the 95% confidence intervals (CI) for their LC_50_ did not overlap [118].

### 4.6. Genetic Linkage Analysis

A genetic linkage experiment was conducted to determine whether the *PxMetAP1* gene was involved in the Bt resistance of *P. xylostella*. The F_1_ progeny were generated by a single-pair cross of the G88 female with the Cry1S1000 male. A single pair of reciprocal crosses between F_1_ and Cry1S1000 produced a back-cross (BC) population (a and b) for a total of 10 families. Back-cross family a (BCa1–5) was produced by an F_1_ female with a Cry1S1000 male, while back-cross family b (BCb1–5) was produced by an F_1_ male with a Cry1S1000 female. The concentration of the Cry1Ac toxin that killed 100% of the F_1_ progeny heterozygotes was determined as the discriminating concentration by toxicity bioassay (see Section 4.2). About 40 3rd instar larvae from each back-cross family were chosen for experiments in the control group (normal diet) and the treatment group (a diet containing a discriminating concentration of Cry1Ac toxin), respectively. All surviving larvae were collected after 72 h, and the genomic DNA (gDNA) of a single larva was extracted using a Blood/Tissue/Cell genomic DNA Extraction Kit (Tiangen, Beijing, China), for genetic linkage analysis between alternative splicing forms of the *PxMetAP1* gene and Cry1Ac resistance. 

A pair of specific primers (see Table 1) were designed based on the principle of the intron length polymorphism of different genotypes of *PxMetAP1* in order to detect the genotypes of *PxMetAP1* in the surviving individuals (R*_M1_* mutation corresponds to the mutation of *PxMetAP1* in Cry1S1000 at the genome level, while that of G88 was S*_M1_*), thus allowing us to determine the relationship between different *PxMetAP1* genotypes and Cry1Ac resistance. PCR amplification was performed using Hieff Canace^®^ High-Fidelity DNA Polymerase (Yeasen, Shanghai, China), and the resistant allele R*_M1_* and the susceptible allele S*_M1_* were distinguished by 2% TAE agarose gel electrophoresis. Fisher’s exact test in the SPSS Statistics 22 software was used to analyze whether the genotype ratio of all surviving individuals in the treatment and control groups in each back-cross family was significantly different and whether the genotype ratio of R*_M1_*R*_M1_*:R*_M1_*S*_M1_* in the control group met the expectation of 50%.

### 4.7. Preparation of sgRNA and Embryo Microinjection

The target selection principle of sgRNA is 5′-GGN19GG-3′ or 5′-CCN19CC-3′, which is transcribed in vitro by the T7 promoter (5′-TAA TAC GAC TCA CTA TAG G-3′), with the last two bases (GG) as the transcription starting point. The two targets (primer sequence shown in Table 1) of *PxMetAP1* are located in exon 2, and the specificity of the targets in *P. xylostella* was determined using an online tool (http://www.rgenome.net/cas-offinder/, accessed on 23 March 2021). The transcription template of sgRNA was synthesized using Hieff Canace^®^ High-Fidelity DNA Polymerase (Yeasen, Shanghai, China) and purified using an EZNA Gel Extraction Kit (Omega, Morgan Hill, GA, USA). All of the components were mixed evenly, according to the instructions of an MEGAscript^TM^ T7 Transcription Kit (Thermo, Waltham, MA, USA), and were placed in an incubator at 37 °C for 16 h for the transcription of sgRNA in vitro. Finally, sgRNA were purified using RNA extraction reagent (Enol:chloroform:isoamylol = 25:24:1; SolarBio, Beijing, China). Cas9 protein (Cas9-N-NLS Nuclease) was purchased from Genscript Biotech Corporation (Genscript, Nanjing, China).

The embryo microinjection system of *P. xylostella* included 1 µL of CAS9-N-NLS Nuclease (final concentration of 300 ng/μL), 1 µL of sgRNA1 (final concentration of 500 ng/μL), 1 µL of sgRNA2 (final concentration of 500 ng/μL), 0.5 µL of 10× Reaction Buffer, and 1.5 µL of RNase-Free ddH_2_O, which were mixed evenly and incubated at 37 °C for 20 min. The needle of a capillary glass tube filled with 2 µL incubation solution was fixed on the microinjector (IM 300, Narishige, Tokyo, Japan), after which the egg card was spread on a glass slide and embryo injection was carried out together with a stereo microscope (SZX16, Olympus, Tokyo, Japan).

### 4.8. Screening and Construction of Homozygous Mutant Strains

F_1_ progeny were generated via single-pair crosses between G88 adults and the surviving adults of the G_0_ generation adults, after which individuals of G_0_ were collected, and single-headed gDNA was extracted using a Blood/Tissue/Cell genomic DNA Extraction Kit (Tiangen, Beijing, China). The primer sequences for specific detection are detailed in Table 1. Hieff Canace^®^ High-Fidelity DNA Polymerase (Yeasen, Shanghai, China) was used to amplify the sequence. Then, the PCR products were sent to Sangon Biotech (Sangon, Xiamen, China) for sequencing, and a mutation was considered to have occurred if a nested peak was observed behind the target position. The progeny of G_0_ with mutations were left, the F_1_ progeny were single-pair crossed with G88 to produce F_2_, and the gDNA of F_1_ continued to be extracted for mutation detection. Afterward, the F_2_ of homozygous mutant individuals were sorted in a single-pair to generate the F_3_, where only the parents with homozygous mutants were retained to continue single mating until a homozygous mutant strain was obtained (Appendix A). Their progeny were propagated to obtain the stable homozygous mutant strains for subsequent experiments.

### 4.9. Activity Assay with Methionine Aminopeptidase (MetAPs)

The activity of the MetAPs of *P. xylostella* was determined according to the previous description of Yang et al. (2012) [119]. Methionine-p-nitroaniline solution is colorless and is hydrolyzed to yellow p-nitroaniline (PNA) and free methionine under the action of MetAPs. PNA has a maximum absorption value at the wavelength of 380 nm. The amount of enzyme required to hydrolyze methionine-p-nitroaniline to produce 1 μmol of PNA per minute at 60 °C at a pH of 8.0 was defined as one unit of enzymatic activity (IU).

First, the 4th instar larvae (n = 10) of the G88 and the *PxMetAP1* homozygous mutant strain were selected and rapidly frozen in liquid nitrogen. Then, 500 µL of ammonia-ammonium chloride buffer solution (Yuanye, Shanghai, China, pH = 8.0, 40 mmol/L) was added to each tube and ground thoroughly on ice using a grinding stick. After that, 500 µL of ammonia-ammonium chloride buffer solution (Yuanye, Shanghai, China, pH = 8.0, 40 mmol/L) was added, and the mixture was shaken evenly, upside-down, and put into a high-speed refrigerated centrifuge (Sorvall LYNX 6000, Thermo Fisher, MA, USA). The supernatant obtained by centrifugation is the crude enzyme solution of the MetAPs, which was stored at −80 °C separately. Then, 6 mmol/L of methionine p-nitroaniline enzyme substrate solution (Taijia, Hangzhou, China) and seven concentration gradients of PNA standard solution (Macklin, Shanghai, China) were prepared. The prepared PNA standard solutions, with gradient concentrations, were then assayed using a microplate reader (Synergy H1, Biotek, Winooski, VT, USA) at a wavelength of 380 nm, using pure water as a control, and each concentration was repeated three times. The standard curve was drawn, with the absorbance of each concentration minus the absorbance of pure water as the ordinate (ΔOD) and the concentration of PNA (μmol/L) as the abscissa.

Finally, 40 µL of ammonia-ammonium chloride buffer solution (Yuanye, Shanghai, China, pH = 8.0, 40 mmol/L) and 20 µL of methionine p-nitroaniline enzyme substrate solution (Taijia, Hangzhou, China) were added to a microwell of the microtiter plate along with 20 µL of diluted crude enzyme solution. The lid was covered, and the plate was incubated for reaction in an electric thermostatic water bath (DK-8D, Yiheng, Shanghai, China) at 60 °C for 10 min, after which 120 µL of stopping solution (36% acetic acid, Macklin, Shanghai, China) was immediately added to stop the reaction. The absorbance was measured at 380 nm using a standard analyzer (Synergy H1, Biotek, Winooski, VT, USA), and the enzyme activity was calculated according to the standard curve. The crude enzyme solution was diluted into three concentrations, and each concentration was repeated three times. For the blank control, the crude enzyme solution was replaced by a buffer solution. The formula for calculating the enzymatic activity is as follows:(1)$$A=\frac{C\times V_{T}\times D}{T\times V_{E}}$$
where:*A*—Enzyme activity, with the unit IU/mL;*C*—The absorption value brought into the standard curve to obtain the concentration of PNA produced by the reaction, with the unit μmol/L;*V_T_*—The total volume of the reaction system in µL;*D*—Dilution of crude enzyme solution;*T*—Reaction time in min; *V_E_*—The volume of diluted crude enzyme solution added in µL.

In our test, *V*_T_ was 200 µL, *T* was 10 min, and *V*_E_ was 20 µL. The independent samples *t*-test (*p* < 0.001) in the SPSS Statistics 22 software was used to analyze the significance of the differences in the enzyme activity determination results between different samples.

## Figures and Tables

**Figure 1 ijms-23-13005-f001:**
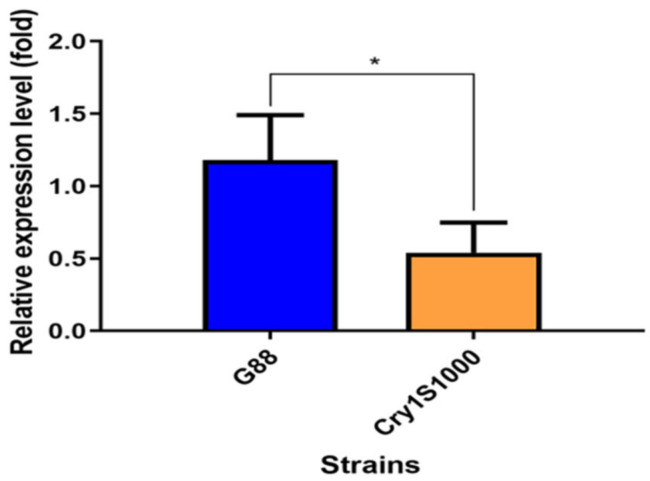
Relative expression level of the *PxMetAP1* gene in the midgut of the 4th instar larvae of the G88 and Cry1S1000 strains. Independent samples *t*-test was used to analyze the differences between the two strains, where * denotes a significant difference (*p* < 0.05).

**Figure 2 ijms-23-13005-f002:**
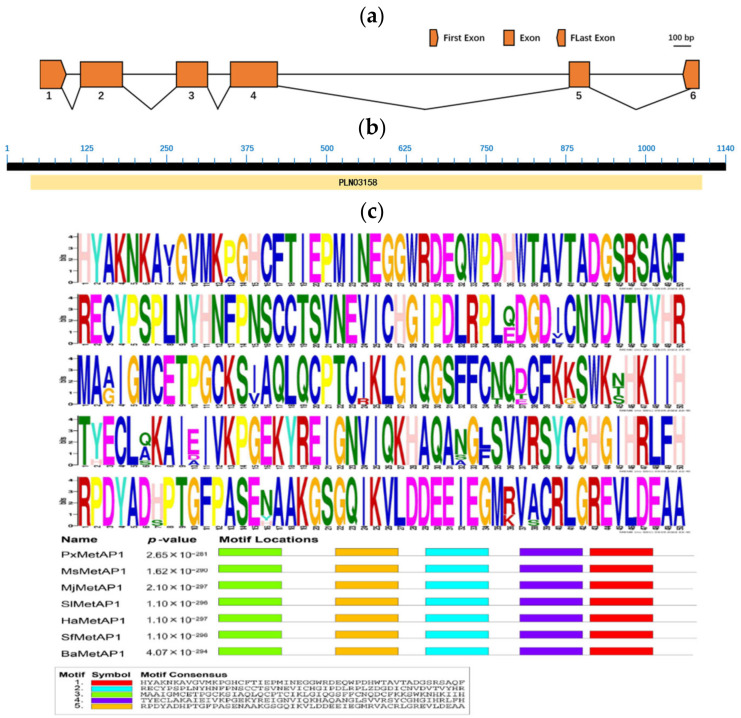
Structure and features of the *PxMetAP1* gene in the G88 strain. (**a**) Genomic structure of the *PxMetAP1* gene in *P. xylostella*. Exons are denoted by orange boxes, and introns are denoted by spaces between two boxes. The scale bar that corresponds to the figure’s scale is displayed; (**b**) annotation of the deduced *PxMetAP1* protein sequence using the NCBI conserved domain database (CDD). The sequence was recognized as a distinctive member of the superfamily of methionine aminopeptidases (MetAPs); (**c**) illustration of the five motifs found in the *PxMetAP1* gene. The figure above shows the amino acid frequency diagram of the *PxMetAP1* gene, while the figure below shows the locations of the motifs in the *PxMetAP1* gene.

**Figure 3 ijms-23-13005-f003:**
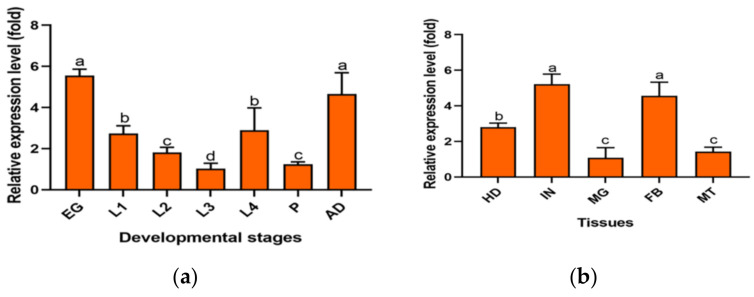
Spatiotemporal expression profiles of the *PxMetAP1* gene in the G88 strain. (**a**) EG, egg; L1, 1st instar larvae; L2, 2nd instar larvae; L3, 3rd instar larvae; L4, 4th instar larvae; P, pupae; AD, adults. (**b**) HD, head; IN, integument; MG, midgut; FB, fatbody; MT, Malpighian tubules. One-way ANOVA was used for the comparison of significant differences in different developmental stages and tissues, and the LSD method was used for multiple comparisons. Different letters denote significant differences (*p* < 0.05).

**Figure 4 ijms-23-13005-f004:**
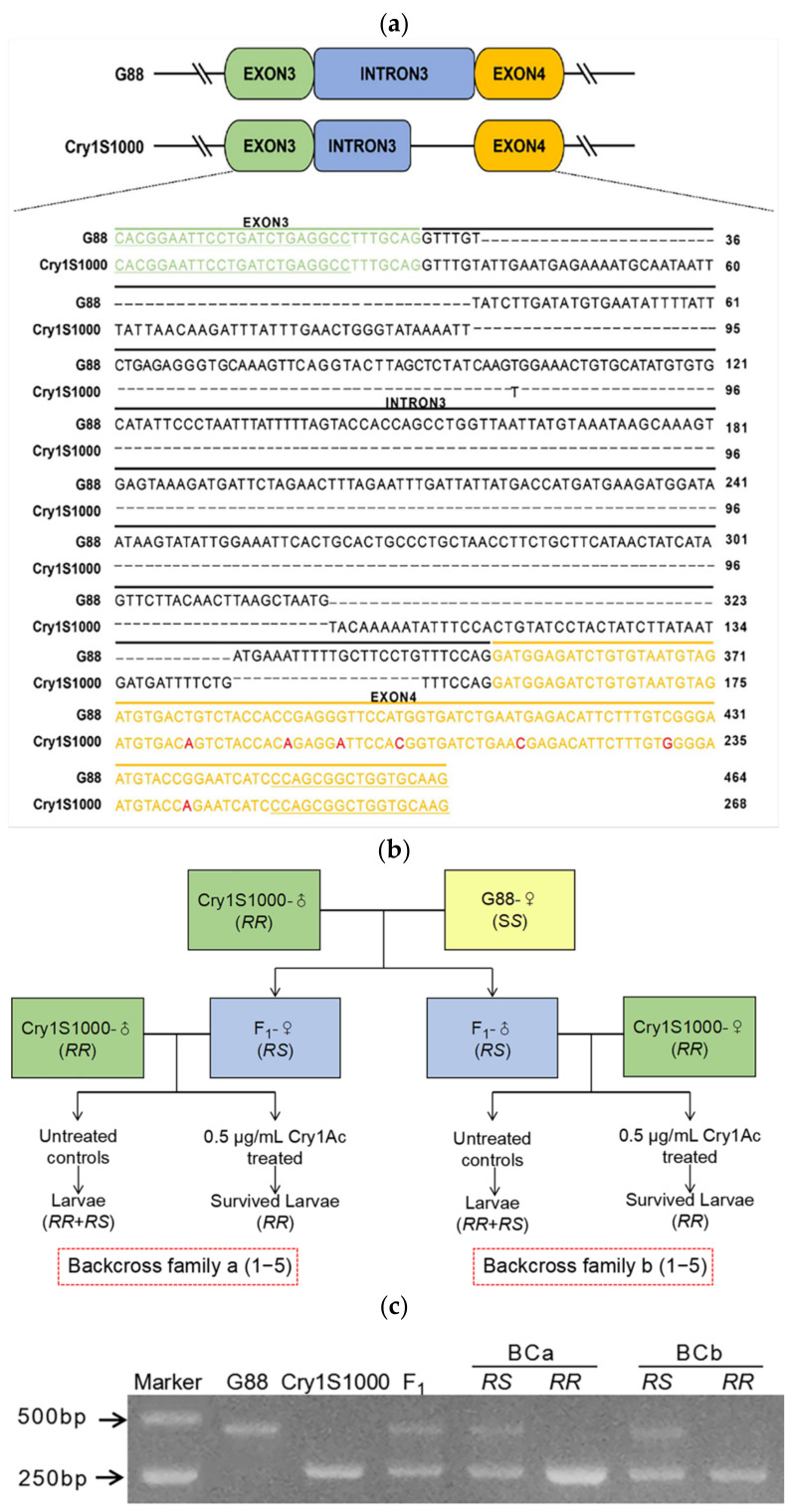
Genetic linkage of *PxMetAP1* with Cry1Ac resistance. (**a**) Allelic sequence alignment of *PxMetAP1* gene in G88 and Cry1S1000 strains. The sequences highlighted in green and orange are the exon 3 and 4 regions, respectively, the underlined sequences are the forward and reverse complementary primers that specifically verify the intron lengths of the different genotypes of *PxMetAP1*, and the red bases represent the point mutation in exon 4. (**b**) Diagram illustrating the strategy for back-crossing and toxicity bioassays. The F_1_ generation is produced by the single-head pairing of G88 (*SS*) female adults and Cry1S1000 (*RR*) male adults. BCa1–5 is produced by the single-pair mating of the F_1_ generation (*RS*) female adults with the Cry1S1000 male adults, and BCb1–5 is produced by the single-pair mating of the F_1_ generation (*RS*) male adults with the Cry1S1000 female adults. The surviving individuals in the control group are *RR* and *RS*, and the surviving individuals in the treatment group (0.5 μg/mL Cry1Ac protoxin) are RR; (**c**) genotypes of the *PxMetAP1* gene in G88 (*S_M1_S_M1_*) and Cry1S1000 (*R_M1_R_M1_*) strains and in the F_1_ generation (*R_M1_S_M1_*) generated by hybridization.

**Figure 5 ijms-23-13005-f005:**
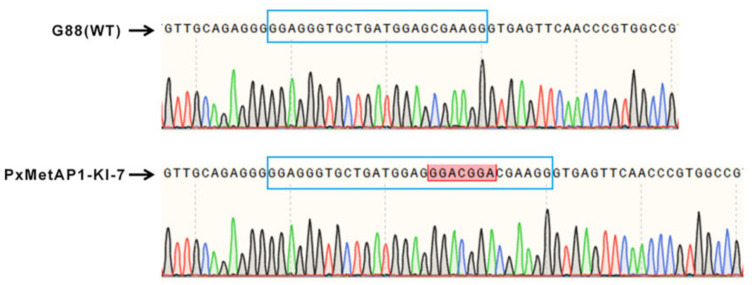
Mutation of the *PxMetAP1* gene in the G88 strain induced by CRISPR/Cas9. In the comparison diagram of bases between the G88 strain and the homozygous mutant strain (PxMetAP1-KI-7), the long blue frame represents the target sgRNA1, the red frame indicates the 7 bp base insertion at this site, the green line represents base A, the red line represents base T, the black line represents base G, and the blue line represents base C.

**Figure 6 ijms-23-13005-f006:**
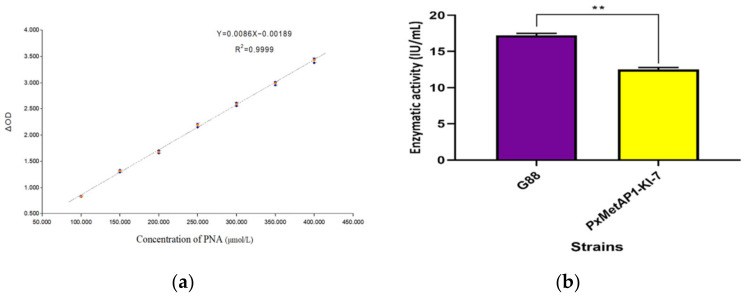
Determination of MetAPs activity of *P. xylostella*. (**a**) Standard curve of p-Nitroaniline; (**b**) results of MetAPs activity determination between G88 strain and homozygous mutant strain (PxMetAP1-KI-7). Independent samples *t*-test was used to analyze the significant difference between the G88 and Cry1S1000 strains; ** denotes an extremely significant difference (*p* < 0.001).

**Figure 7 ijms-23-13005-f007:**
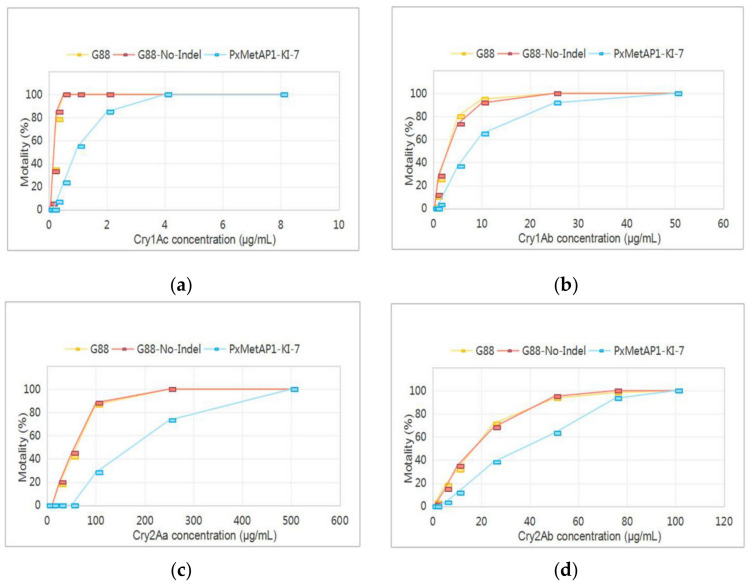
Reaction of different *P. xylostella* strains to Bt toxins. (**a**) Cry1Ac toxin; (**b**) Cry1Ab toxin; (**c**) Cry2Aa toxin; and (**d**) Cry2Ab toxin.

**Table 1 ijms-23-13005-t001:** The list of all primers of the *PxMetAP1* gene in this research.

Primer	Sequence (5′→3′)
MetAP1-CDS-F1	ATGGCCGGAATTGGAATGTGCG
MetAP1-CDS-R1	TCATCGAGCACCTCGCGTC
MetAP1-CDS-F2	GCCTGGTCCGAAGCGTACG
MetAP1-CDS-R2	CTTGCACCAGCCGCTGG
MetAP1-CDS-F3	CACGGAATTCCTGATCTGAGGCC
MetAP1-CDS-R3	CTAGGAGTTTAGCTTCTCCATTTGGTCC
Q-MetAP1-F	AGCTCGGCATACAAGGCTCG
Q-MetAP1-R	TGTGAACCCGTACGACGGCC
RPL32-F	CCAATTTACCGCCCTACC
RPL32-R	TACCCTGTTGTCAATACCTCT
MetAP1-GL-F	CACGGAATTCCTGATCTGAGGCC
MetAP1-GL-R	CTTGCACCAGCCGCTGG
MetAP1-sgRNA1-F	TAATACGACTCACTATAGGAGGGTGCTGATGGAGCGAGTTTTAGAGCTAGAAATAGCAAGTTAA
MetAP1-sgRNA2-F	TAATACGACTCACTATAGGCCGACGCCGGGAACCCGGTCGTTTTAGAGCTAGAAATAGCAAGTTAA
MetAP1-sgRNA-R	AGCACCGACTCGGTGCCACTTTTTCAAGTTGATAACGGACTAGCCTTATTTTAACTTGCTATTTCTAGCT

Note: The target sequence of sgRNA is marked with double underline.

## Data Availability

The datasets generated for this study are available on request from the corresponding author.

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
