# Peer review of "The Potential Role of the Methionine Aminopeptidase Gene PxMetAP1 in a Cosmopolitan Pest for Bacillus thuringiensis Toxin Tolerance"

_ijms, 2022, doi:10.3390/ijms232113005_

Round 1

Reviewer 1 Report

This manuscript is well written and minor revisions are suggested, as follow:

1. Sub headings in the results and materials and methods should be consistence with similar order. This would be easier for the readers to follow by comparing the methods and results of each experiment.

2. Some figures may need more description to make those are able to stand alone; e.g. additional of the strain used for comparing different life stages and origins of the tissues.

3. Discussion on why the expression the gene of interest in the fourth instar was higher than that in the first to the third instar?

4. The relative expression of L4 in Fig 1 and Fig 3 was different (ca 1 vs 3), why?

Details comments and suggestions can be found in the attached file.

Reviewer 2 Report

I think the manuscript is interesting and that some researchers in the field may be interested to take a look on. I think the most relevant aspect of the article is the conclusion that the PxMetAP1 gene product is implicated in the tolerance of P. xylostella to Bt toxins.
